# Nonlinear Hydraulic Pressure Response of an Improved Fiber Tip Interferometric High-Pressure Sensor

**DOI:** 10.3390/s20092548

**Published:** 2020-04-30

**Authors:** Wei Huang, Zhe Zhang, Jun He, Bin Du, Changrui Liao, Shen Liu, Guolu Yin, Yiping Wang

**Affiliations:** 1Guangdong and Hong Kong Joint Research Centre for Optical Fibre Sensors, College of Physics and Optoelectronic Engineering, Shenzhen University, Shenzhen 518060, China; huangwei2017@email.szu.edu.cn (W.H.); 2150190115@email.szu.edu.cn (Z.Z.); dubin2016@email.szu.edu.cn (B.D.); cliao@szu.edu.cn (C.L.); shenliu@szu.edu.cn (S.L.); glyin@szu.edu.cn (G.Y.); ypwang@szu.edu.cn (Y.W.); 2Guangdong Laboratory of Artificial Intelligence and Digital Economy (SZ), Shenzhen University, Shenzhen 518060, China; 3Key Laboratory of Optoelectronic Devices and Systems of Ministry of Education and Guangdong Province, Shenzhen University, Shenzhen 518060, China

**Keywords:** hydraulic pressure, nonlinear, Fabry–Perot interferometer

## Abstract

We demonstrate a silica diaphragm-based fiber tip Fabry–Perot interferometer (FPI) for high-pressure (40 MPa) sensing. By using a fiber tip polishing technique, the thickness of the silica diaphragm could be precisely controlled and the pressure sensitivity of the fabricated FPI sensor was enhanced significantly by reducing the diaphragm thickness; however, the relationship between the pressure sensitivity and diaphragm thickness is not linear. A high sensitivity of −1.436 nm/MPa and a linearity of 0.99124 in hydraulic pressure range of 0 to 40 MPa were demonstrated for a sensor with a diaphragm thickness of 4.63 μm. The achieved sensitivity was about one order of magnitude higher than the previous results reported on similar fiber tip FPI sensors in the same pressure measurement range. Sensors with a thinner silica diaphragm (i.e., 4.01 and 2.09 μm) rendered further increased hydraulic pressure sensitivity, but yield a significant nonlinear response. Two geometric models and a finite element method (FEM) were carried out to explain the nonlinear response. The simulation results indicated the formation of cambered internal silica surface during the arc discharge process in the fiber tip FPI sensor fabrication.

## 1. Introduction

In-fiber Fabry–Perot interferometric (FPI) pressure sensors have been widely used in various industrial fields owning to their outstanding advantages such as light weight, miniature size, immunity to electromagnetic interference, and survivability in harsh environments [1,2,3,4]. For example, in underground or undersea oil/gas exploration engineering, where pressure measurement was performed in high-temperature, high-pressure, corrosive, flammable, and explosive environments [5,6,7], FPI pressure sensors exhibit high sensitivity, excellent safety, and robustness. As a reflective fiber sensor, the FPI pressure sensor can perform probe measurements, which is more flexible and convenient. Driven by insistent demands in practical applications, many methods have been developed for fabricating FPIs, such as femtosecond laser micromachining [8,9], chemical etching [10,11], or even by simply cleaving and splicing a piece of photonic crystal fiber [12,13]. Among them, fiber tip elastic diaphragm Fabry–Perot interferometers (FPIs) are especially good candidates for probe-type measurements. Various materials such as polymers [14,15], metal [16], silica [3,17], or even two-dimensional nanomaterials such as graphene [18] have been reported for constructing the fiber tip diaphragm. In 2017, Zhang et al. proposed a fiber tip PVC-diaphragm FPI pressure sensor [15]. The sensor was fabricated by welding a piece of PVC diaphragm to the end facet of a well-cut single-mode fiber. The production method is simple and the sensor exhibits a high pressure sensitivity of 65.5 nm/MPa. However, polymers diaphragm FPI pressure sensors exhibited poor thermal resistance and suffered a large temperature drift. Metal materials such as silver FPI pressure sensors showed an improved thermal resistance compared with polymer diaphragm FPI pressure sensors. In 2012, Xu et al. demonstrated a fiber tip nanothick silver diaphragm FPI pressure sensor [16]. The nanothick silver diaphragm FPI exhibits a higher pressure sensitivity and a better thermal stability than the polymer-based FPI. However, the sensor cannot survive in a high-pressure environment due to the fragility of the silver diaphragm. Moreover, the transfer process of silver to the fiber tip is complicated. The graphene diaphragm has very high mechanical strength and can be stretched by as much as 20%. In 2012, Ma et al. developed a fiber tip graphene diaphragm FPI pressure sensor [18]. This FPI sensor shows an extremely high pressure sensitivity and can even be employed for acoustic pressure sensing [19]. However, the graphene diaphragm is fragile and the production repeatability of this type of FPI sensor is poor.

Alternatively, the silica diaphragms are considered to be the most robust and temperature- resistant choice for fiber tip FPI sensors. Many literatures have reported fiber tip silica diaphragm pressure sensors until now [3,4,16,20,21]. For example, in 2011, Ma et al. demonstrated a fiber tip all-silica FPI high-pressure sensor. The sensor can operate at a wide pressure range of 0 to 40 MPa [20]. However, the sensitivity of this type of sensor is low, which may be attributed to the spherical air cavity of the FPI. In 2017, Liu et al. demonstrated a rectangular air cavity FPI pressure sensor by employing a fusion diaphragm transfer method [22]. Due to the rectangular air-cavity shape and nanosilica diaphragm, the sensor exhibited a high pressure sensitivity of 12.22 nm/kPa. However, the pressure measurement range of the sensor was limited to 0 to 30 kPa and the high-pressure performance of the sensor was not explored or exhibited. As far as we are concerned, the fiber tip FPI pressure sensors either had low sensitivity or small pressure measurement range, and the highest pressure sensitivity achieved in a pressure range of 0 to 40 MPa is 315 pm/MPa [20].

In this paper, a high sensitivity (−1.436 nm/MPa) and large measurement range (0–40 MPa) pressure sensor based on a fiber tip all-silica diaphragm FPI is experimentally demonstrated. By employing the proposed fiber tip polishing technique, the thickness of silica diaphragm can be precisely controlled and the pressure sensitivity of the FPI sensor was improved significantly with the reduction of diaphragm thickness, while the relationship is not linear. Nonlinear response of the sensor to high hydraulic pressure was experimentally observed in case the silica diaphragm is thin enough (in our case, 2.09 μm). Geometric modeling and numerical simulation based on the proposed sensor structure indicated the formation of fiber tip cambered silica diaphragm, which may be resulted from the arc discharge process during FPI fabrication. Detailed analysis in both experiments and simulations was carried out to reveal the relationship between the sensitivity, linearity, and diaphragm thickness. The proposed FPI pressure sensor has the advantages of robust structure, high sensitivity, and large measurement range, which may find many applications in petroleum exploration industry. Moreover, the FPI sensor is suitable for extensive production due to the simplicity and consistence of the fabrication method.

## 2. Sensor Fabrication and Working Principle

The schematic diagram and optical microscope image of the proposed sensor structure are shown in Figure 1a,b, respectively. At first, a section of hollow-core fiber (HCF) (INNOSEP-TSP07515, CN) with an inner diameter of 75 μm was fusion spliced with a lead-in standard single-mode fiber (SMF) (Corning“SMF-28e”, Wilmington, NC, USA), the core diameter of which is ~8.2 μm. The outer diameter (i.e., 125 μm) of the HCF was the same as the lead-in SMF. Then, the HCF was cut off by a homemade fiber cutter with an accuracy of ±5 μm under a microscope [23]. The length of remained HCF was several hundred micrometers. After that, the remained HCF was fusion spliced with a piece of coreless fiber (CF) (Thorlabs “FG125LA”, Morganville, NJ, USA), followed by cutting off the CF pigtail. Thus, a fiber tip air cavity FPI was formed and the CF acts as the diaphragm of FPI. Subsequently, the fiber tip FPI was placed vertically into a fiber lensing machine (ULTRAPOL), in which the CF was polished and the diaphragm thickness was gradually reduced. The equipment and detailed device fabrication process are similar to that reported in our previous work [21]. By optimizing the polishing parameters, such as grit size of polishing paper (1 µm), rotating speed (100 r/min), and polishing time (~15 min), six sensor samples (i.e., S1–S6) with fiber tip diaphragm thickness (d) of 10.20, 7.78, 6.02, 4.63, 4.01, and 2.09 μm, respectively, were produced using this method. Such a thin silica diaphragm in fiber tip FPIs is attractive for highly sensitive pressure measurements.

Figure 2a–f shows the reflection spectra of the six fabricated FPI sensors (i.e., S1–S6) and the corresponding scanning electron microscope (SEM) images of these fiber tip silica diaphragms created by a desktop Scanning Electron Microscope (Phenom Pharos, Eindhoven, NL), respectively. The side view SEM images of silica diaphragms were obtained by cutting open the FPI air cavity by use of a femtosecond laser. The reflection spectra of S1–S6 were obtained by connecting the sensor pigtail with a broadband optical source (BBS) (Fiber-Lake “FL-ASE-EB”, Shenzhen, CN) and an optical spectrum analyzer (OSA) (Yokogawa” AQ6370C”, Shanghai, China) through a 3 dB fiber coupler. As can be clearly seen in Figure 2, all the reflection spectra of S1–S6 could be characterized by dense interference fringes modulated by big envelops, corresponding to the optical interference signals formed by the air cavity and the silica diaphragm, respectively.

The reflection intensity can be expressed as [21]
(1)$$\begin{array}{l} I={\left|E\right|}^{2}={\left|E_{1}-E_{2}\exp\left[j\left(\frac{4\pi }{\lambda }n_{air}L\right)\right]+E_{3}\exp\left[j\left(\frac{4\pi }{\lambda }\left(n_{air}L+n_{silica}d\right)\right)\right]\right|}^{2}\\ \approx E_{1}{}^{2}+E_{2}{}^{2}+E_{3}{}^{2}-2\left(E_{1}E_{2}-E_{1}E_{3}\right)\cos\left(\frac{4\pi }{\lambda }n_{air}L\right)-2E_{2}E_{3}\cos\left(\frac{4\pi }{\lambda }n_{silica}d\right)\;(d\to 0)\;, \end{array}$$
where *E*_1_, *E*_2_, and *E*_3_ are the amplitudes of the three reflection waves; *L* is the length of air cavity; *d* is the thickness of fiber tip silica diaphragm; and *λ*, *n_ai_*_r_, and *n_silica_* denote the light wavelength and refractive index (RI) of air and silica, respectively. From Equation (1) we can see that, when the silica diaphragm is thin enough (i.e., *d*→0), the reflection intensity is only determined by the phase item $4\pi n_{air}L/\lambda$. The dip wavelength (i.e., wavelength at intensity minimum) in the reflection spectrum can be deduced by
(2)$$\lambda_{\min}=\frac{4n_{air}L}{2m+1},$$
where *m* is a positive integer. When the pressure was applied to the FPI sensor, the silica diaphragm will be deformed due to the pressure difference between inside and outside of the cavity. The length of air cavity will change from *L* to *L +* Δ*L* when pressure was applied. The change of cavity length will result in a wavelength shift Δ*λ* according to Equation (2)
(3)$$\Delta \lambda =\frac{4n_{air}}{2m+1}\Delta L,$$

Here, the RI of air was assumed to be constant since the little deformation of silica diaphragm results in little change of air cavity volume. The pressure sensitivity of a spherical cavity FPI pressure sensor can be expressed as [15]
(4)$$S=\frac{\delta \lambda }{\delta P}=\frac{\delta \lambda }{\delta L}\frac{\delta L}{\delta P}=\frac{4n_{air}}{2m+1}\frac{(1-\nu )R^{2}}{2Ed},$$
where P, *ν*, *E*, and *R* represent the applied pressure, Poisson’s ratio, Young’s modulus of silica, and the radius of the spherical cavity, respectively. In our case, the radius, R, is approximately infinite as the diaphragm is flat. Therefore, the pressure sensitivity of the FPI sensor can be improved by decreasing the diaphragm thickness. This is the basic working principle of the proposed sensor.

## 3. High Hydraulic Pressure Response

A hydraulic pressure test equipment, schematically illustrated in Figure 3, was employed to study the high pressure response of six sensor samples, i.e., S1 (*d* = 10.20 μm), S2 (*d* = 7.78 μm), S3 (*d* = 6.02 μm), S4 (*d* = 4.63 μm), S5 (*d* = 4.01 μm), and S6 (*d* = 2.09 μm). The hydraulic pressure test equipment consists of a hydraulic machine (ConST181A) and a digital pressure gauge (ConST211), which are incorporated in a hydraulic chamber to generate and display real-time pressure in the chamber. The sensor head is placed into the hydraulic chamber through a feed-through of the chamber followed by sealing the feed-through by employing a strong glue. The pigtail of the sensor sample was connected with a broadband optical source (BBS) and an optical spectrum analyzer (OSA) through a fiber 3 dB coupler. The applied pressure was increased from 0 to 40 MPa in increments of 2 MPa. At each measuring point, the sensor was kept for 5 min for obtaining a stable spectrum. One of the dip wavelengths around 1550 nm was traced with increasing pressure. The recorded dip wavelength with respect to the applied pressure for the six sensor samples, S1–S6, is shown in Figure 4a, respectively. We can clearly see that with the decreasing of *d*, pressure sensitivity was improved accordingly while the linearity of pressure response decreased significantly as well. In other words, the linear response range of the sensor was narrowed down with the decreasing of d. This can be attributed to the elastic and plastic deformation property of the silica diaphragm. When the applied pressure is beyond the threshold of elastic deformation of the silica diaphragm, plastic deformation dominates. The plastic deformation of silica diaphragm will result in an uneven change of cavity length of the FPI, i.e., the nonlinear pressure response of the sensor. As clearly observed in Figure 4a, the pressure response of S5 (*d* = 4.01 μm) and S6 (*d* = 2.09 μm) in the range of 0 to 40 MPa shows a parabolic shape, whereas pressure response of S1 (*d* = 10.20 μm), S2 (*d* = 7.78 μm), S3 (*d* = 6.02 μm), and S4 (*d* = 4.63 μm) showed a good linearity. The linear fitting of the pressure response of S1–S4 yields a high correlation coefficient of 0.999, 0.999, 0.999, and 0.991 and pressure sensitivity of −0.173, −0.243, −0.466, and −1.436 nm/MPa, respectively. Figure 4b illustrates the pressure sensitivities of the six sensor samples S1–S6 at pressures of 10, 20, 30, and 40 MPa, respectively. An interesting finding is that the pressure sensitivity is not reversely proportional to the d. As can be clearly seen in Figure 4b, the sensor S4 (*d* = 4.63 μm) exhibits the highest sensitivity at 40 MPa. The observation of this phenomenon is not occasional. Many other samples have been produced and employed for the pressure tests and similar phenomena were observed. This indicates an underlying physical mechanism involved in the sensor structure. Detailed modeling and numerical simulation were performed in the next section and a cambered silica diaphragm model agrees well with the experiment results.

Measurement repeatability is one of the most vital indicators for a pressure sensor. An experiment to verify the sensor’ measurement repeatability was conducted by long-term and repeat cycle tests. The sensor samples S4 (*d* = 4.63 μm) and S6 (*d* = 2.09 μm) were employed, and the pressure response were studied by the same test equipment described above. The applied pressure was increased from 0 to 40 MPa in a step of 2 MPa. The reflection spectra were recorded after maintaining the pressure value for 5 min. When the applied pressure reached 40 MPa, the sensor samples were held at 40 MPa for 12 h to verify the sensors’ high-pressure resistance. Then, the applied pressure was decreased from 40 to 0 MPa in a step of 2 MPa by the same procedure. Reflection spectra of the sensor samples S4 (*d* = 4.63 μm) and S6 (d = 2.09 μm) were recorded by the same method described above. This procedure is defined as a test cycle. Three test cycles were implemented for the two sensor samples S4 (*d* = 4.63 μm) and S6 (*d* = 2.09 μm), respectively. Figure 5 shows the experimental results, where an error bar (at *Y* axis) was employed to characterize the maximum measuring error in different measurement cycles. As is clearly depicted in Figure 5, pressure response of sensor sample S4 (*d* = 4.63 μm) shows a good linearity and smaller measuring error (Figure 5a), while pressure response of sensor sample S6 (*d* = 2.09 μm) exhibits a much poorer linearity and larger measuring error (Figure 5b). Moreover, the measurement error of S6 (*d* = 2.09 μm) is more significant at the low pressure with a range of 0 to 5 MPa and is several times larger than in 5 to 40 MPa range and the slop of curve (i.e., pressure sensitivity) flattens as the pressure increased. The inset of Figure 5a,b shows the pressure sensitivities of the two FPI sensors S4 (*d* = 4.63 μm) and S6 (*d* = 2.09 μm) at different pressure values, respectively. Another interesting finding is that the sensitivity of S6 (*d* = 2.09 μm) reaches its maximum of −9.76 nm/MPa when the applied pressure is ~2 MPa, as clearly shown in the inset of Figure 5b. Various factors may be account for the phenomenon. On the one hand, the silica diaphragm will transform from elastic to plastic deformation with pressure increasing. Moreover, on the other hand, we suspect that the silica diaphragm of the produced fiber tip FPI may be a slight cambered surface other than a standard plane due to the arc discharge fusion process. Numerical simulations for a plane and slight cambered surface were carried out respectively in the following section. The calculation results agreed well with our conjecture.

## 4. Theoretical Modeling and Discussion

For a deep understanding of the observed experimental phenomena, the finite element method (FEM) and a commercial simulation software “COMSOL” were employed to model the sensor structure. Two geometric models, i.e., plane surface and cambered surface silica diaphragm with curvature radius *R*, were established as shown in Figure 6a,b, respectively. First, Model 1 (plane silica diaphragm) was employed to calculate the deformation of six sensor samples S1 (*d* = 10.20 μm), S2 (*d* = 7.78 μm), S3 (*d* = 6.02 μm), S4 (*d* = 4.63 μm), S5 (*d* = 4.01 μm), and S6 (*d* = 2.09 μm) under different pressures, respectively. The deformation of silica diaphragm was then converted into the wavelength shift of the interference spectra according to Equation (3). The calculation results for S1–S6 are plotted in Figure 7a–f, respectively. In Figure 7, the black square dot (■) denoted the calculated results according to Model 1, and the red dots (●) denoted the experimental results. It can be clearly seen that the deviations between the calculation and experiment for S1–S4 were small, while for S5–S6, the deviations were large. Then, Model 2 (cambered surface silica diaphragm with curvature radius *R*) was carried out to calculate the deformation of silica diaphragm. The deformation of silica diaphragms with different thickness and curvatures were calculated and compared, respectively. Figure 8a shows the calculated sensitivity of the six samples S1–S6 with the same diaphragm curvature radius *R* of 400 μm, respectively. Figure 8b shows the calculated pressure sensitivities of S6 with different diaphragm curvature radius *R* ranging from 100 to 500 μm, respectively. The pressure sensitivity increases drastically and the pressure value of the maximum sensitivity shifts to lower pressures in case the d is reduced (thinner diaphragm, Figure 8a) or the diaphragm curvature radius *R* is increased (flatter diaphragm, Figure 8b). This can be understood by the fact that, when the cambered diaphragm turns symmetric under a proper pressure, the sensitivity of sensor reaches maximum, a thinner or flatter diaphragm renders a lower pressure for the cambered diaphragm to turn to symmetric. The tendency of sensitivity curve agrees well with the experiment results that shown in the inset of Figure 5b. The curvature radius of 400 μm was considered to be the best value that matches the experimental results best. For comparison, the calculation results were plotted in Figure 7a–f, respectively, denoted by blue triangle dots (▲). The calculation results agreed well with the experimental results. Based on the geometric model and calculation results, we believed that, during the fabrication of the sensor, a cambered silica surface was formed due to the arc discharge during welding. It is worth noting that, the resulted curvature of cambered surface is small and when the diaphragm is thick enough, the resulted cambered surface of diaphragm is not obvious. With the thick decreasing of diaphragm, the effects of cambered surface become more and more obvious in the pressure tests. The proposed geometric model and simulation results agree well with the observed experimental phenomena.

## 5. Conclusions

We demonstrated a high-sensitivity, high-pressure sensor based on fiber tip all-silica FPI. The fiber tip polishing technique was employed to reduce the thickness of silica diaphragm and the pressure sensitivity is improved significantly with the reduction of d. A high sensitivity (−1.436 nm/MPa) and good linearity (0.991) in a wide pressure range of 0 to 40 MPa were experimental demonstrated for an FPI sensor with a diaphragm thickness (*d*) of 4.63 μm. The achieved pressure sensitivity and measurement range are of great improvement compared with previously reported fiber tip FPI pressure sensors. A thinner silica diaphragm, i.e., 4.01 and 2.09 μm, renders a higher pressure sensitivity of −1.472 and −4.689 nm/MPa, respectively. This means the d can be reduced further according to the required pressure range, and hence it provides a flexible design for various applications. Moreover, nonlinear hydraulic pressure response of the proposed FPI sensor was found in the experiments when the diaphragm is thin enough. Geometric modeling and theoretical calculations indicates the formation of a cambered surface diaphragm, which may be attributed to the arc discharge fusion process during sensor fabrication. Additionally, the simplification and consistence of the fabrication method make the proposed sensor promising for extensive production.

## Figures and Tables

**Figure 1 sensors-20-02548-f001:**
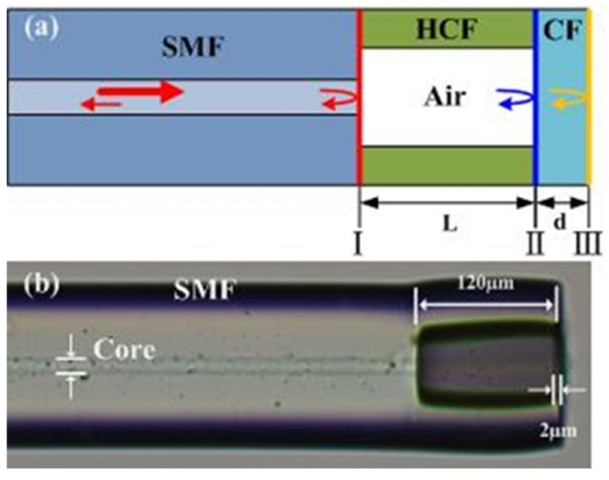
(**a**) Schematic and (**b**) microscope images of the proposed fiber tip all-silica Fabry–Perot interferometer (FPI) sensor.

**Figure 2 sensors-20-02548-f002:**
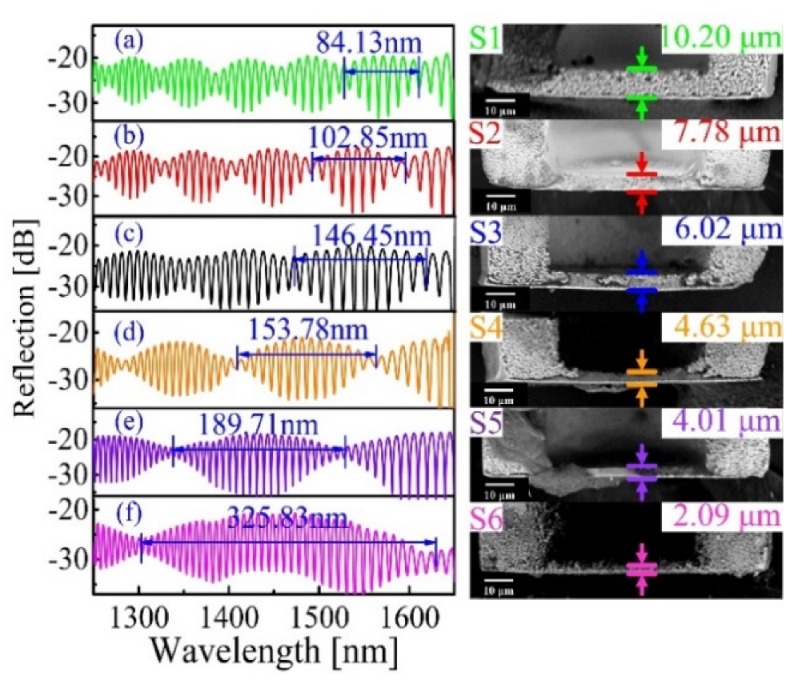
Reflection spectra (left column) and the corresponding SEM images (right column) of the six fabricated fiber tip FPI sensors with different diagram thickness. (**a**) S1 (*d* = 10.20 μm), (**b**) S2 (*d* = 7.78 μm), (**c**) S3 (*d* = 6.02 μm), (**d**) S4 (*d* = 4.63 μm), (**e**) S5 (*d* = 4.01 μm), and (**f**) S6 (*d* = 2.09 μm).

**Figure 3 sensors-20-02548-f003:**
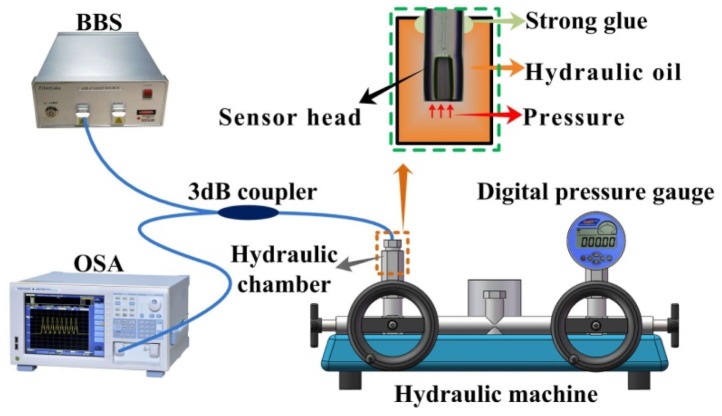
Hydraulic pressure test equipment.

**Figure 4 sensors-20-02548-f004:**
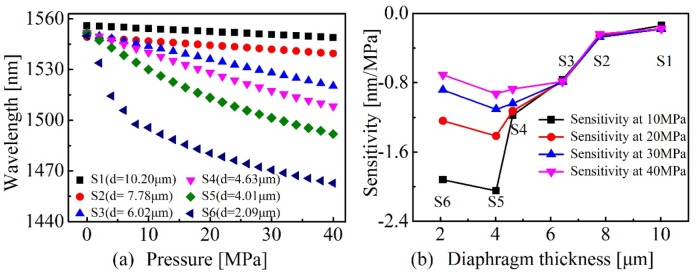
(**a**) The tracked dip wavelength versus applied pressure for six samples: S1 (*d* = 10.20 μm), S2 (*d* = 7.78 μm), S3 (*d* = 6.02 μm), S4 (*d* = 4.63 μm), S5 (*d* = 4.01 μm), and S6 (*d* = 2.09 μm), respectively. (**b**) Pressure sensitivities of the six sensor samples at pressures of 10, 20, 30, and 40 MPa, respectively.

**Figure 5 sensors-20-02548-f005:**
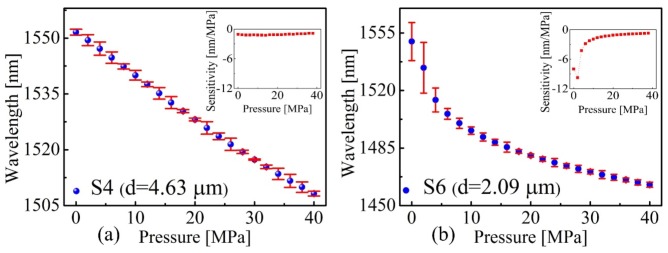
Wavelength of the tracked dip versus pressure for (**a**) S4 and (**b**) S6, respectively. The insets show the corresponding pressure sensitivities of S4 and S6 as functions of pressure.

**Figure 6 sensors-20-02548-f006:**
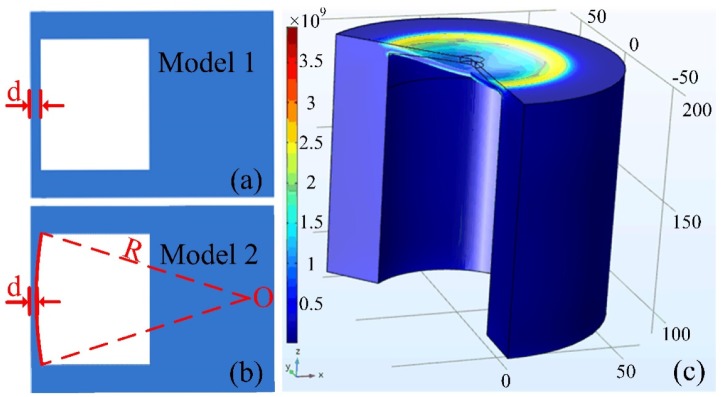
Schematics of the established geometric models: (**a**) Model 1 (i.e., plane silica diaphragm) and (**b**) Model 2 (i.e., cambered surface silica diaphragm with curvature radius *R*). (**c**) Simulated deformation of the diaphragm under pressure.

**Figure 7 sensors-20-02548-f007:**
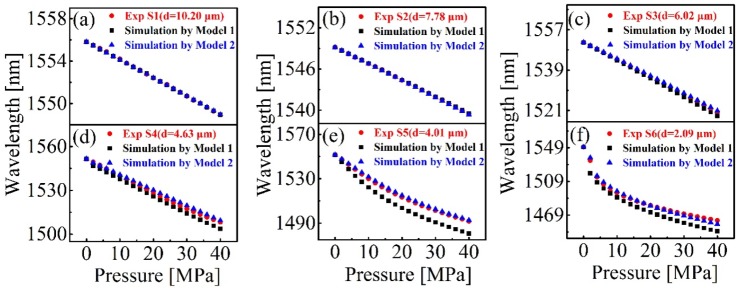
Experimental (red dots) and calculation results of the six sensor samples: (**a**) S1 (*d* = 10.20 μm), (**b**) S2 (*d* = 7.78 μm), (**c**) S3 (*d* = 6.02 μm), (**d**) S4 (*d* = 4.63 μm), (**e**) S5 (*d* = 4.01 μm), and (**f**) S6 (*d* = 2.09 μm). (The black square dots denote the calculation results by Model 1 and the blue triangle dots represent the calculation results by Model 2 with curvature radius of 400 μm.)

**Figure 8 sensors-20-02548-f008:**
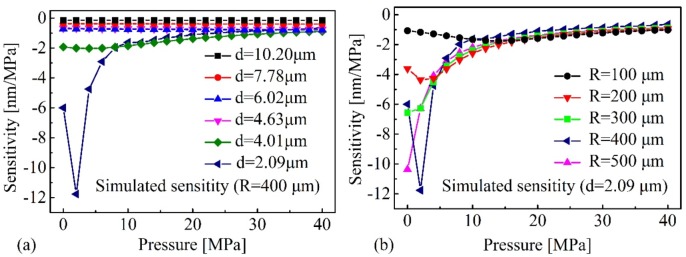
(**a**) Calculated pressure sensitivities of the six sensor samples S1–S6 by Model 2 with the same diaphragm curvature radius *R* of 400 μm. (**b**) Calculated pressure sensitivities of S6 (*d* = 2.09 μm) with diaphragm curvature radius *R* ranging from 100 to 500 μm.

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
