# Peer review of "Nonlinear Hydraulic Pressure Response of an Improved Fiber Tip Interferometric High-Pressure Sensor"

_sensors, 2020, doi:10.3390/s20092548_

Round 1

Reviewer 1 Report

Dear editors,

Fiber-based high-pressure sensors are attractive and useful in many applications, for example, in the underground or undersea oil/gas exploration industry. In this paper, the authors present a silica diaphragm-based fiber-tip Fabry-Perot interferometer (FPI) for measuring hydraulic pressure of up to 40 MPa. By optimizing the diaphragm thickness of FPI, high sensitivity, good linearity and repeatability are demonstrated on the FPI high-pressure sensors. Moreover, the device fabrication process for the sensor probe is quite simple, which is promising for practical use.

In my point of view, the idea proposed in this manuscript is adequately novel. These results are quite interesting and also convincing. However, some important technique issues still need further explanation. As a result, I recommend the manuscript could be published on Sensors after some minor revisions.

Detailed comments and suggestions are described as follows:

  1. In Section 2 “sensor fabrication and working principle”, the authors said “by optimizing the polishing parameters, such as grit size of polishing paper, rotating speed and polishing time, six sensor samples (i.e. S1-S6) with fiber-tip diaphragm thickness was produced using this method.” I suggest the authors should give more detailed information on these parameters. These factors can determine the performance of these sensor samples.

  1. In Fig. 5, the measurement repeatability has been evaluated by error bar in Y axis. However, the fabrication repeatability is another important factor for practical use. Could the authors give more information on the fabrication repeatability?

  1. In Fig. 8, it seems a resonant point exists on the hydraulic pressure sensitivity under different pressures. The resonant pressure sensitivity increases drastically and the resonant point shifts to lower pressures in case the diaphragm thickness is reduced (thinner diaphragm, Fig. 8(a)) or the diaphragm curvature radius R is increased (flatter diaphragm, Fig. 8(b)). I suggest the authors could give more explanations on this issue.

  1. The English of this manuscript should be further improved. For example, there are even some grammar mistakes, such as “With the decreasing of diaphragm thickness, the effects of cambered surface on the deformation of diaphragm under pressure becomes more and more prominent.” and “The observed experimental phenomenon were well explained by the proposed model and calculations.”

Reviewer 2 Report

Dear Authors,

you demonstrated a silica diaphragm-based fiber-tip Fabry-Perot interferometer (FPI) for measuring hydraulic pressure.  The motivation of the study seems to be well grounded. The article includes appropriate references.

I think that the manuscript is not suitable for publication in this form. Below is a list with comments:

  • The construction of the article is very illegible, e.g. Figure 2 should be in the results, and modeling should be after the introduction. I am asking for systematizing paper. Please use the magazine's guidelines.
  • Please add more information about equipment used in the experiment: optical fiber, light source, optical spectrum, analyser. Please add the city and county of the production of the devices.
  • Please improve the quality of all figures.
  • „The length of remained HCF was several hundred micrometers”. How much? Why such a length?
  • The diameter of the SMA fiber core should be given if you specify the inner diameter HCF.
  • Does the roughness of silica diaphragm-based affect measurements? If so, which one?
  • Please add references to the formulas and explain all used variables.
  • The number after the dot should be standardized.
  • You must determine if you are using the name: optical fiber or fiber.
  • Are there single-use sensors? if so, why can't they be used repeatedly?
  • How does it combine individual elements? What method do you use? What equipment do you use to fusion splice?
  • Why the equipment by which SEM images were created was not described? This should be added. Why is there no scale on SEM images?
  • Why do the d values ​​in the text are different from the values ​​in the SEM images? You should systematize it.
  • How did you choose the signal envelopes? Is the period of change is constant for the diaphragm-based?
  • You should not write membrane thickness values (d) everywhere. This is tiring for the reader.
  • Please correct the unit's record with the value in [] on the axes.
  • Please change the title. In this form it is not encouraging to read.
  • The manuscript needs an English correction.

In conclusion, the manuscript should undergo a substantial revision before being resubmitted.

Reviewer 3 Report

The manuscript entitled “Nonlinear Hydraulic Pressure Response of Fiber-Tip Interferometric High-Pressure Sensor with Enhanced Sensitivity” presents a Fabry-Perot interferometer design for pressure measurement. Similar designs have already been reported by many researchers. However, the authors claim that they have higher pressure measurement range with better sensitivity, which is interesting. The design is simple and has easier fabrication. Having said that the reviewer has many technical issues with the manuscript and all of them must be explained/corrected before it could be further considered for publication.

  1. The author experimented with six sensors with different silica diaphragm thickness. In line 168, contrary to theoretical understanding, the author concludes that pressure sensitivity is not inversely proportional to the diaphragm thickness. The fact is that this relationship is not showing in the experimental results means that the design is not perfect. Please talk about design flaws and their effects.

  1. The author emphasizes strongly on the cambered silica diaphragm for the non-linear performance of the sensors which have diaphragm thickness less than 4.63 microns. The cambered silica diaphragm is just one of the possibilities. Another factor could be the imperfect splicing or between HCF and CF fibers. Imperfect splicing could include angle between the splicing surfaces, the air bubble in the splicing etc. One other possible factor could be reaching the elastic limit of silica at high pressure in case of thin diaphragms. Have these factors been investigated?

  1. The experimental data for sensor S2 and S3 in Figure 4(a) are not at all consistent with the data in Figure 7(b) and 7(c). The wavelength measurement seems more or less for S2 and S3 the same in Figure 4(a) while they do not match in Figure 7(b) and 7(c).

  1. In line 183, the author claims that the performance of S4 is linear, however, the data for S4 in Figure 4(a) and 4(b) suggests otherwise. How can it be claimed as a linear response when the sensitivity of S4 is changing drastically with pressure as clearly seen in Figure 4b.

  1. The data for S4 shown in Figure 4(a) and Figure 5(a) are unacceptably different. The wavelength at 40MPa in Figure 4a is shown to be less than 1495nm and in Figure 5(a) is more than 1505nm and the error bar at 40MPa is very small in Figure 5(a). Similarly, the pressure sensitivities for S4 is clearly nonlinear in Figure 4(a) and almost linear in Figure 5a. How is S4 data in Figure 4(a) is not in congruence with the repeatability test data in Figure 5(a)?

  1. Apart from cambered diaphragm the non-linearity for sensor S6 could be because of the in-elastic deformation in the diaphragm as its thickness is too small and at high pressure is could surpass its elastic limit.

The concerns related to the above points 3, 4 and 5 are very serious. These point to serious flaws and raise serious doubts. Please explain them in detail. Other important issues are as follows

  1. There are only 5 graphs in Figure 8a.

  1. The abstract is too long, please try to keep it more concise.

9. The author should focus on the technical correctness of the manuscript. In line 54, the sentence ‘However, polymer exhibited poor …………. drift’ should start as ‘However, the metal diaphragm FPI pressure sensors exhibited poor …….. drift’.  In line 66, the sentence ‘Many literatures ….. ’ should be paraphrased. Auxiliary verbs should be used carefully i.e. ‘is’ should be used in line 129 instead of ‘was’.

Reviewer 4 Report

This paper presents a FP sensor of a larger pressure measurement range and describes why the sensitivity changes with the pressure. Several questions should be clarified before publication.

(1) The manufacture process of the FP sensor should be clear. What is the type of fiber used to form FP and what is the type of machine the authors use to complete the manufacture?

(2) The theory section. The sensitivity changes with a large pressure may be caused by the approximate model of the circular diaphragm with a little bending. The author should compare the model of circular diaphragm bending when the pressure is little and large with COMSOL finite element method. The model has indicated that the sensitivity is nonlinear with different pressure scale. 

(3) Experimental section. The spectrum of this FP sensor should be clear with different pressure under test. Besides, what is the FSR and fineness of this FP sensor? What is the reflectivity of the two reflector surface? 

Reviewer 5 Report

The paper under evaluation is devoted to the fabrication and characterization of a silica diaphragm-based fiber-tip Fabry-Perot interferometer (FPI) devoted to hydraulic pressure measurements. The paper is well organized, and clear in the text and figures. The authors demonstrate high sensitivity (-1.436 nm/MPa) and good linearity (0.991) in a wide pressure range between 0 - 40 MPa for one of the sensor under study. The experimental founds are well supported by theoretical and modelling analysis. In my opinion the paper is robust and can be published in the present form. Only one minor minor comment:

Section 2, line 92, please define the microscope. Is it optical microscope?

Round 2

Reviewer 2 Report

Dear Authors,

The revision you performed increased the quality of the proposed paper. All the remarks and comments I raised have been satisfactorily addressed. In my opinion, the manuscript after the revision is suitable for publication, making a good contribution to the field. 

Author Response

Thanks a lot for your comments and suggestions.

Reviewer 3 Report

The author’s efforts to clear serious doubts are appreciated. There are still some issues to be addressed  

  1. The sensitivity is clearly increasing with reducing thickness of the diaphragm in Figure 4(b). However, the relationship is not linear. This point must be corrected by highlighting these two experimental facts throughout the manuscript.
  2. In line 22 of the abstract, the words ‘with a decreasing’ should be replaced with ‘by reducing’. The sentence ‘The thinner……..response ’ in line 26 should be rewritten. The sentence in line 183-184 should be rewritten. The language at many places in the manuscript is not up to the mark. The author is strongly advised to read the entire manuscript carefully and make amendments wherever necessary.
